# Systematic Assessment of Small RNA Profiling in Human Extracellular Vesicles

**DOI:** 10.3390/cancers15133446

**Published:** 2023-06-30

**Authors:** Jing Wang, Hua-Chang Chen, Quanhu Sheng, T. Renee Dawson, Robert J. Coffey, James G. Patton, Alissa M. Weaver, Yu Shyr, Qi Liu

**Affiliations:** 1Department of Biostatistics, Vanderbilt University Medical Center, Nashville, TN 37232, USA; jing.wang.1@vumc.org (J.W.); hua-chang.chen@vumc.org (H.-C.C.); quanhu.sheng.1@vumc.org (Q.S.); 2Center for Quantitative Sciences, Vanderbilt University Medical Center, Nashville, TN 37232, USA; 3Department of Cell and Developmental Biology, Vanderbilt University School of Medicine, Nashville, TN 37232, USA; renee.dawson@vanderbilt.edu (T.R.D.); robert.coffey@vanderbilt.edu (R.J.C.); alissa.weaver@vanderbilt.edu (A.M.W.); 4Center for Extracellular Vesicle Research, Vanderbilt University School of Medicine, Nashville, TN 37232, USA; 5Department of Medicine, Vanderbilt University Medical Center, Nashville, TN 37232, USA; 6Department of Biological Sciences, Vanderbilt University, Nashville, TN 37232, USA; james.g.patton@vanderbilt.edu; 7Department of Pathology, Microbiology and Immunology, Vanderbilt University Medical Center, Nashville, TN 37232, USA

**Keywords:** extracellular vesicles, small RNA-seq, quality control, technical biases, RNA composition, RNA biotypes enrichment, small RNA profiling of EVs

## Abstract

**Simple Summary:**

Small RNA sequencing has been widely used for characterizing the landscape of small non-coding RNAs—the most abundant cargo in extracellular vesicles (EVs). Here, we performed a systematic assessment of the quality, technical, and potential biological biases introduced by different EV isolation methods and the enrichment of specific, small RNA biotypes in EVs. The findings in this study guide the quality control of EV small RNA-seq and the selection of EV isolation techniques and enhance the interpretation of small RNA contents and the preferential loading of specific RNA biotypes into EVs.

**Abstract:**

Motivation: Extracellular vesicles (EVs) are produced and released by most cells and are now recognized to play a role in intercellular communication through the delivery of molecular cargo, including proteins, lipids, and RNA. Small RNA sequencing (small RNA-seq) has been widely used to characterize the small RNA content in EVs. However, there is a lack of a systematic assessment of the quality, technical biases, RNA composition, and RNA biotypes enrichment for small RNA profiling of EVs across cell types, biofluids, and conditions. Methods: We collected and reanalyzed small RNA-seq datasets for 2756 samples from 83 studies involving 55 with EVs only and 28 with both EVs and matched donor cells. We assessed their quality by the total number of reads after adapter trimming, the overall alignment rate to the host and non-host genomes, and the proportional abundance of total small RNA and specific biotypes, such as miRNA, tRNA, rRNA, and Y RNA. Results: We found that EV extraction methods varied in their reproducibility in isolating small RNAs, with effects on small RNA composition. Comparing proportional abundances of RNA biotypes between EVs and matched donor cells, we discovered that rRNA and tRNA fragments were relatively enriched, but miRNAs and snoRNA were depleted in EVs. Except for the export of eight miRNAs being context-independent, the selective release of most miRNAs into EVs was study-specific. Conclusion: This work guides quality control and the selection of EV isolation methods and enhances the interpretation of small RNA contents and preferential loading in EVs.

## 1. Introduction

Extracellular vesicles (EVs) are lipid-bound particles that are secreted from cells into the extracellular environment. Although EVs were initially considered to be part of a waste-removal mechanism to discard unwanted cellular materials [1], increasing evidence suggests that EVs play a fundamental and evolutionarily conserved role in cellular communication by delivering molecular cargo [2,3,4,5,6,7,8,9]. EVs carry a range of molecular cargo, including lipids, DNA, RNA, and proteins, which can be taken up by recipient cells and alter their gene expression and function [8,10,11]. Decoding EV cargo is important since cargo composition reflects the functional state of donor cells, and the alteration of cargo is associated with homeostasis and disease progression [12,13,14,15,16].

Among the variety of molecular cargo in EVs, small non-coding RNAs are the most abundant, including miRNAs, rRNA fragments, tRNA fragments, Y RNAs, snRNAs, and snoRNAs [1,17]. Among these, miRNAs are the best characterized in post-transcriptional alteration of gene expression in recipient cells, affecting cellular responses to stress and inflammation and driving disease progression [18]. As an example, miR-155 in adipocyte-derived microvesicles mediates M1 macrophage polarization, which reciprocally regulates insulin signaling and glucose uptake in adipocytes and thus causes chronic inflammation and local insulin resistance [19,20]. In addition, small non-coding RNAs in EVs can serve as biomarkers and therapeutic targets in a variety of diseases, including HIV, cancer, cardiovascular disease, Type 1 and Type 2 Diabetes, COVID-19, glioma, and neurological disorders [21,22,23,24,25,26,27]. miR-486-3p is a good example of a circulating biomarker, as it has been reported to distinguish glioblastoma from lower-grade astrocytoma [28]. Therapeutically, the use of EVs to deliver miR-219a-5p might be a feasible and promising strategy to induce remyelination in multiple sclerosis patients, since the overexpression of miR-219a-5p in EVs induced more differentiation of oligodendrocyte precursor cells than liposomes and polymeric nanoparticles [29]. These findings highlight the need for new technologies for the association of biomarkers with a specific exosome subtype and the exosome subtype to a particular function and/or group of functions [26,27,30].

Small RNA sequencing has been widely used for characterizing small RNA landscapes in EVs [29,31,32,33,34,35,36,37,38,39,40,41,42,43]. Depending on the size of the vesicle and the purification strategy, some studies have revealed a large overlap and good correlation between EV and cellular small RNA, whereas other studies have demonstrated selective miRNA exports [34,35,36,37,40,42,43]. Although these findings shed light on small RNAs in EVs, they are limited to certain cell types or conditions. Several EV repositories have been developed, including exoRBase [44], EVmiRNA [45], and EVAtlas [46], which provide rich resources of long RNA and small RNA profiling in EVs. However, there is a lack of systematic analyses to provide a global picture of the quality and preferential loadings of small RNAs in EVs. The Extracellular RNA Communication Consortium (ERCC) led an effort to build a reference catalog of extracellular RNA in five human biofluids covering twenty-three health conditions [47]. However, the samples were collected primarily from cell-free biofluids and contained combinations of vesicular and non-vesicular particles, which resulted in significant variability between and within studies [48]. Here, we focused on small RNAs within membrane-bound EVs. We collected and reanalyzed all the publicly available EV small RNA-seq datasets from biofluids, cell lines, and primary cell cultures. We performed a systematic assessment of the quality, technical, and potential biological biases introduced by different EV isolation methods and specific biotype enrichment in EVs compared to matched cellular levels. Our findings not only guide quality control and the selection of EV isolation methods but also enhance the interpretation of small RNA contents and preferential loading of specific RNA biotypes from individual studies as well as across studies.

## 2. Materials and Methods

### 2.1. Data Sources

We used the terms “(exosome OR exosomes OR ectosomes OR microvesicles OR microvesicles OR “EV” OR “extracellular RNA” OR (extracellular AND vesicle) OR (extracellular AND vesicles)) AND “Homo sapiens”[porgn] AND “gse”[Filter]) NOT “exosome complex” NOT “nuclear exosome” NOT “RNA exosome” AND “high throughput sequencing”[PTYP]“ to search the Gene Expression Omnibus (GEO) and the Sequence Read Archive (SRA). In addition, we reviewed all datasets deposited in the ERCC and publications listed on vendor websites selling exosome extraction kits. We downloaded and manually checked all publicly available datasets from the GEO, SRA, and ERCC. We only kept small RNA-seq datasets from human EVs and removed those from non-vesicles, non-small RNA-seq, or other organisms. To ensure EVs are enriched, we only kept studies where EVs were isolated using either commercial exosome extraction kits, such as Exo-Quick, ExoEasy, qEV, and total exosome isolation kit (Thermo, Waltham, MA, USA), or differential ultracentrifugation, which have been proven to isolate high-yield and high-purity EVs [43,49,50,51,52]. Besides, most studies further verified the presence of EVs using several methods, including Western blotting of EV-enriched proteins (such as CD63, CD81, and CD9) [53,54,55,56,57] and non-EV-enriched proteins [53,55,58,59,60], qualitative (such as electron microscopy and atomic force microscopy) and quantitative methods (such as nanoparticle tracking analysis, dynamic light scattering, tunable resistive pulse sensing, and high-resolution flow cytometry) [53,55,57,58,59,61], and a wide-field and a close-up electron microscopy image [55,61,62]. Notably, each EV isolation method is prone to isolating some contaminants. This study analyzes the presence of different small RNA species in different EV-enriched isolates (or fractions) from different methods. After filtering, 2756 samples from 83 studies including 55 studies with EVs only and 28 studies with both EVs and matched donor cells, were kept for downstream analysis. We pulled experimental metadata from the original source and then conducted manual curation to try to standardize the information, such as extraction methods and small RNA preparation kits.

### 2.2. Small RNA-Seq Data Analysis

We processed and analyzed all the raw sequencing data uniformly. Since a variety of commercial and customized small RNA library preparation kits were used, different adapter patterns were included in raw reads because adaptor sequences are not always available. To streamline the analysis, we developed a Python package, FindAdapt, for fast, accurate, and automatic detection of adapter patterns without any prior information (https://github.com/chc-code/findadapt, accessed on 23 October 2022). The adapter patterns identified by FindAdapt, including adapter sequences and random bases at 5′ and 3′ ends, were provided to our standard small RNA-seq analysis pipeline, TIGER, for adapter trimming, reads mapping to host and non-host genomes, and quantification and differential expression of a variety of small RNA biotypes [63]. Briefly, Cutadapt (v2.10) [64] was used to trim 3′ adapters and random 5′ and 3′ bases. Reads with fewer than 16 nucleotides were designated as “too short” and discarded. Quality control on both raw reads and adaptor-trimmed reads was evaluated using FastQC (v0.11.9) (www.bioinformatics.babraham.ac.uk/projects/fastqc (accessed on 23 October 2022)). After trimming, reads were first mapped to a customized database built from the host genome (GENCODE GRCh37.p13) and from known sequences of host mature transcripts curated in specific library databases (such as for miRNAs in miRbase and tRNAs in GtRNAdb2) by Bowtie1 (v1.3.0) [65], allowing one mismatch. Mapped reads were assigned to different classes of annotated small RNAs, including miRNA, tRNA, rRNA, snRNA, snoRNA, Y RNA, and lincRNA. Unmapped reads longer than 19 nucleotides were then aligned to non-host genomes in parallel, including exogenous structural RNA databases and curated exogenous genome databases (bacteria, fungal, algae, and viral), allowing no mismatches. Raw counts and normalized counts per million total counts (CPM) were reported for each small RNA.

### 2.3. Small RNA-Seq Data Normalization and Comparison

The proportion of total mapped reads and host genome reads was calculated by the number of total mapped reads and host genome reads divided by the total number of reads after trimming. Small RNA abundance was determined by the number of small RNA reads divided by the number of host genome reads. The abundance of each type of small RNA was determined by the number of reads that mapped to that type of small RNA relative to the total number of small RNA reads. To reduce the potential bias from characteristic compositions of different EV isolation methods, three-way ANOVA with the EV isolation method as a confounding variable was used to estimate the significance of enrichment or depletion of RNA biotypes in EV compared to cells. The expression of miRNAs was normalized to the median value across all samples to identify highly expressed miRNAs in both EVs and cells, as well as EV-specific miRNAs. For each study with matched cell and EV samples, DESeq2 [66] was used to detect differentially expressed miRNAs with the EV isolation method as a confounding variable. miRNAs with FDR < 0.05 and an absolute value of fold change > 1.5 were selected to be significantly differential.

## 3. Results

### 3.1. A Global View of Quality in EV Small RNA Sequencing

After excluding 513 samples from the donor cells, we assessed the quality of 2243 EV datasets using the total number of reads after adapter trimming, the overall mapping rates to both the host and non-host genomes, the total number of reads and the mapping rates to the host genome only, and the proportion of small RNA and miRNA reads (Appendix A). After adapter trimming, most (94.6%) datasets had more than 100,000 reads and greater than 20% mapping rates when aligned to both host and non-host genomes (Figure 1A, Appendix A). Data with less than 100,000 reads or 20% mapping rates were flagged and excluded from downstream analyses. Most of the excluded samples came from GSE100467 (38 samples), GSE148576 (38 samples), and GSE115572 (24 samples), all of which were large-scale studies (>100 samples) from blood or plasma. Since other samples from the same experimental series achieved a decent number of reads and mapping rates, caution needs to be taken for analyzing and interpreting those samples with extremely low numbers of reads and mapping rates. Besides examining the overall mapping rates to both host and non-host genomes, it is important to consider the number and the percentage of reads aligned to the host genome only. The ERC Consortium quality control for small RNA-seq data requires a minimum of 100,000 reads mapped to the host genome, and the percentage of the host genome reads greater than 50% [48]. Among the 2120 datasets, 1963 datasets had more than 100,000 reads mapped to the host genome; 1681 contained greater than 50% of host genome reads, and 78.6% of the datasets met both criteria (Figure 1B, Appendix A). In addition, we further evaluated the percentage of small RNA and miRNA in the host genome reads (Figure 1C,D). As expected, most host genome reads were mapped to small RNAs. Small RNAs accounted for greater than 75% of the host genome reads in 93.8% of the datasets (Figure 1C). In comparison, miRNAs showed a wide range of distribution across the datasets, with some having a very low fraction of miRNAs (<10%), while others contained a high percentage of miRNA reads (>80%) (Figure 1D). Overall, 78.5% of datasets had miRNA reads constituting greater than 10% of the host genome reads (Figure 1D, Appendix A). We decided to exclude the experimental series where more than half of the samples had either <75% small RNAs or <10% miRNA reads since they were outliers with high sparsity in miRNAs and other non-coding RNAs.

We further explored the median and the interquartile range (IQR) of small RNA proportions within each study. The IQR is a measure of variability, and a high IQR suggests high variability in the study. Most studies obtained high median values (>0.9) and small IQRs (<0.06), indicating high abundances and low variability in small RNA content (Appendix A). Several studies, however, showed a high IQR in small RNA content (Appendix A). If high variability arose partly from replicates in these studies, caution and additional evaluation are needed to reanalyze and reuse these studies due to low consistency and replicability. After grouping studies by the donor source, we found that EVs from biofluids had significantly higher median values of small RNA proportions than those from cell lines and primary cell cultures (Appendix A, *t*-test: *p* = 2.37 × 10^−5^/0.02). EVs from urine obtained a larger IQR in small RNA proportions than EVs from other biofluids (Appendix A, *t*-test: *p* = 0.003).

There are six commonly used extraction methods: Exo-Quick, ExoEasy, ExoRNeasy, qEV, total exosome isolation kit (Thermo), and differential ultracentrifugation. Sorting studies by EV extraction methods, we found that differential ultracentrifugation generally resulted in significantly larger IQRs in small RNA proportions than other methods, except total exosome isolation kit (Figure 2A, *t*-test: *p* = 0.006/0.002/0.03/0.008/0.26, differential ultracentrifugation compared to Exo-Quick kit, ExoEasy, ExoRNeasy, qEV, and total exosome isolation kit, respectively). To reduce the potential bias introduced by different experimental conditions and small RNA extraction methods, we focused on studies of plasma only and further narrowed down to those using miRNeasy kits for RNA purification. Comparing the variability after restricting to these studies resulted in the same conclusion: that differential ultracentrifugation purification consistently showed higher variability and lower replicability in small RNA proportions within each study (Figure 2B,C). These findings suggest that differential ultracentrifugation may not be as reproducible as other methods for small RNA isolation, which is consistent with previous findings [67,68]. Besides high variability in small RNA proportions within each study, differential ultracentrifugation showed high variability across studies as well, as indicated by the high variation of median values (Figure 2). The high variability across studies might be due to differences in ultracentrifugation equipment, protocols, centrifugation rotors, times, speeds, and whether iodixanol cushions were used [69,70]. 

### 3.2. Small RNA Composition in EVs

There are a variety of small RNA biotypes found in EVs [23,47,71]. We investigated the relative abundance of eight small RNA biotypes across EV studies from plasma, including miRNAs, tRNA fragments, rRNA fragments, Y RNAs, snRNA fragments, snoRNA fragments, mt-tRNA fragments, and miscellaneous RNAs (misc-RNAs). MiRNAs were the most abundant RNA in plasma EVs, with a median proportion value of 39.6%. Besides miRNAs, tRNA fragments, rRNA fragments, and Y RNAs also contributed to a significant portion of small RNAs (Figure 3A and Appendix A). Y RNAs were the second most abundant small RNA (median: 15.9%), followed by rRNA fragments (median: 10.5%) and tRNA fragments (median: 3.2%). In comparison, the proportions of snRNA fragments, snoRNA fragments, mt-tRNA fragments, and misc-RNAs were very low (<1%) (Appendix A). The relative representation of these different small RNAs was highly variable across different EV samples (Appendix A). Interestingly, different EV extraction methods seemed to capture different small RNA compositions. This is consistent with previous studies, which reported that different EV extraction methods have characteristic compositions, that is, EVs with different degrees of other small RNA carriers [8,10,72]. Differential ultracentrifugation seemed to capture more misc-RNAs and snoRNA fragments (Appendix A). In plasma EV samples, Exo-Quick and ExoRNeasy kits captured more miRNAs compared to rRNA fragments (Figure 3B and Appendix A). Eight plasma studies using the Exo-Quick kit and three studies using ExoRNeasy all showed median values of log2-transformed ratios of miRNA/rRNA greater than 1 (Figure 3B). This is consistent with a recent study that ranked Exo-Quick as the top/second for detecting the EV miRNA markers [73]. ExoRNeasy and the total exosome isolation kit seemed to enrich for Y RNAs compared to tRNA fragments (Figure 3C and Appendix A). Six studies using the two kits all showed high Y RNA/tRNA fragment ratios (Figure 3C).

### 3.3. Enrichment of RNA Biotypes in EVs

We analyzed 28 studies that profiled small RNAs from both EVs and their donor cells, which allowed us to explore the preferential enrichment of small RNA biotypes in EVs. Of the 28 studies, we further focused on 15 studies with EVs released from cell lines because they were more biologically homogenous compared to biofluids and primary cell cultures. From these studies, we observed a consistent RNA enrichment preference regardless of EV isolation methods (Figure 4). The overall miRNA proportion in EV was significantly lower than their matched cellular levels (*p* = 7.56 × 10^−64^, three-way ANOVA with the EV isolation method as a confounding variable; Figure 4A). In addition, the miRNA content across these EV samples was generally more variable than that observed in the donor cells. Similarly, snoRNA fragments were detected at significantly lower abundance in EVs compared to matched cellular levels, except for two studies (GSE143613 and GSE85761) (*p* = 4.19 × 10^−21^, three-way ANOVA; Figure 4B). These findings suggest that miRNAs and snoRNA fragments are more likely to be retained in cells than transported into EVs. In contrast, rRNA fragments and tRNA fragments showed significantly higher levels in EVs compared to their cellular levels (*p* = 1.58 × 10^−30^ and *p* = 4.67 × 10^−25^, respectively, three-way ANOVA; Figure 4C,D), suggesting they are more likely to be transported into EVs. 

Focusing on specific miRNAs, we found that 15 miRNAs were abundant in both EVs and donor cells (Figure 5A), including the hsa-let-7 family, miR-26a-5p, miR-30d-5p, miR-25-3p, and miR-21-5p. It is well known that let-7 family members are the most abundant among all miRNAs in the cell [74,75,76], and we found that these miRNAs were highly abundant in EVs as well. In addition, we discovered nine miRNAs that were highly abundant in EVs but not in the cells, including hsa-miR-451a, hsa-miR-486-5p, hsa-miR-122-5p, hsa-miR-146a-5p, hsa-miR-199a-3p; hsa-miR-199b-3p, hsa-miR-143-3p, hsa-miR-144-3p, hsa-miR-21-5p, and hsa-miR-223-5p (Figure 5A). Differential expression with the EV isolation method as a confounding variable also demonstrated that the nine miRNAs were significantly enriched in EVs compared to the cells (log_2_FC > 3 and FDR < 0.01), indicating that their high abundance was not likely to be biased by different EV isolation methods. Among the nine miRNAs, mir-451a and miR-144 have been previously reported to show higher levels in human EVs compared to cellular levels [35].

We calculated the differential expression of miRNAs between EVs and matched cellular levels. Eight miRNAs were found to be preferentially exported into EVs in a context-independent way (Figure 5B highlighted in cyan), and ten other miRNAs were more enriched in EVs in all except three datasets (GSE165323, GSE143613, and GSE85761) (Figure 5B highlighted in magenta). These miRNAs include hsa-miR-199a-3p; hsa-miR-199b-3p, and miR-320 family (miR-320a-3p, miR-320b, miR-320c, and miR-320d). Previously, miR-320b was found to be abundant in exosomes and underrepresented in matched donor cells [43]. Our findings validated that report and suggested that sorting of the miR-320 family members into EVs might be more general. In contrast, seven miRNAs were enriched in cells compared to matched EVs (Figure 5B, highlighted in blue). Although several miRNAs showed an overall preference to be either loaded into EVs or retained in cells, the selective sorting of miRNAs into EVs was mostly study-specific. For instance, seven miRNAs were enriched only in EVs derived from breast cancer cell lines and adipose MSCs (Figure 5B, highlighted in green), one of which was miR-100-5p. miR-100-5p has been reported to inhibit the proliferation, migration, and invasion of breast cancer cells [77]. Five miRNAs including miR-93-5p, miR-125a-5p, miR-1306-5p, miR-147b-3p, and miR-190a-5p were selectively retained in cells only in colorectal cancer cell lines (Figure 5B, highlighted in lime). All of them were abnormally expressed in colorectal cancer [78,79,80,81,82,83]. Selective sorting of functionally important miRNAs into EVs derived from specific cancer cell types might be part of molecular signaling pathways that drive cancer initiation/progression.

## 4. Discussion

An increasing number of studies on EV small RNA profiling have now been published across different cell types and conditions. Here, we provide a comprehensive assessment and comparison of all the publicly available datasets across 83 studies involving a variety of diseases and conditions. We analyzed the overall quality, including the total number of reads, the overall mapping rates to both host and non-host genomes, as well as the mapping rates to the host genome only, and the small RNA and miRNA proportions. The majority of small RNA reads in EVs were derived from miRNAs, Y RNAs, and fragments of rRNA and tRNA. Full-length miRNAs were present in most samples but contributed only a small portion of the overall reads and are highly variable across studies. We also evaluated the technical and potential biological biases introduced by different EV isolation methods. Notably, we found that differential ultracentrifugation had high variability within each study and across studies on small RNA proportions. Together, the analysis of such a large-scale collection provides a global picture of quality in EV small RNA-seq experiments and helps with future quality control and selection of EV isolation methods. 

EV isolates have characteristic compositions. In other words, they are usually enriched in EV and to a different degree in other small RNA carriers that vary according to the isolation principle/method used [8,10,72]. Each of the common EV isolation methods is prone to isolating some contaminants; for example, Exo-Quick and total exosome isolation are likely to obtain proteins by PEG-precipitation, while SEC from plasma usually includes a certain proportion of lipoproteins [72,84,85]. For differential ultracentrifugation, the greater variability in reproducibility of isolating small RNAs could come from the fact that ultracentrifugation succeeds in purifying the EV better than other methods [86]. Also, ultracentrifugation has more equipment, protocol, and operator-dependent variation, hence the variability [10]. Although our analysis tried to reduce the bias from different EV isolation methods, we still need to be cautious about the results interpretation. 

Our study showed low variability in the total small RNA proportions but high variations in the small RNA compositions. The high variability of small RNA compositions across studies might be driven by multiple factors, including experimental conditions, technical factors such as isolation methods, and inherent EV heterogeneity [47,87,88,89]. Given these confounding factors, it is challenging to characterize cargo within specific EVs because of differences in the biogenesis of different classes of EVs due to heterogeneity in size and composition [90]. EVs can be classified by size into exosomes (30–150 nm), microvesicles (100–1000 nm), large oncosomes (1000–10,000 nm), and apoptotic bodies (100–5000 nm), but different studies have used different size ranges in their definition of these particles [91,92]. To make things worse, there is no standard terminology for EV types. Some studies use “exosome” as a generic descriptor of EVs, while others define “exosome” to be EVs strictly originating within the endosomal system [93]. This is compounded by the fact that different names have been used even when the same isolation method was used. For example, some studies using Exo-Quick kits refer to the resulting particles as EVs, while others refer to these particles as exosomes [94,95,96,97,98,99,100,101]. The International Society of Extracellular Vesicles (ISEVs) has recommended standardized nomenclature in which “exosomes” are only used when derived from multivesicular bodies as part of the endosomal system, whereas EVs should be used when the mode of biogenesis is unclear [102,103]. For ongoing retrospective analyses of EVs, adherence to the ISEV standards and methodology is recommended to reduce sources of pre-analytical variability and allow proper comparison between studies [104]. 

In addition to pre-analytical variability, there are also downstream analytical biases driven by the choice of computational methods and assumptions, for example, how to normalize data for accurate comparison. A common assumption in most normalization methods is that the amount of total input RNA content is the same across experimental conditions. However, when comparing small RNA biotypes between cells and EVs, that assumption might not always be true. For example, Sork et al. (2018) found that parental cells expressed abundant levels of small RNA, while the small RNA content in EVs was modest and highly variable across samples [35]. If the total small RNA content is significantly different across conditions, normalization methods could introduce computational bias and lead to wrong conclusions. A way around this problem is to use spike-in controls to create a standard baseline measurement, which would enable accurate measurement of biological differences in total small RNA content between samples.

Despite the challenges above, we sought to determine global enrichment for each small RNA biotype. We found that miRNAs and snoRNA fragments are generally more likely to be retained in cells rather than exported into EVs, while rRNA fragments and tRNA fragments are more likely to be released into EVs. This finding may provide a clue toward understanding the selective sorting of small RNAs into EVs. miRNAs are typically associated with Ago2 within RISC complexes, but eukaryotic cells express multiple Ago proteins. Differential association with Ago proteins, as is observed in plants [105,106], could regulate differential export. Here, we identified several miRNAs exported under most conditions, whereas most miRNAs display context-dependent sorting. One possibility for context-dependent sorting is differential association with specific RNA-binding proteins, which are able to enter EVs and potentially carry their RNA cargo [90,107,108,109,110,111,112,113]. Integration of proteomic and small RNA profiling will help uncover the relationships between specific RBPs and exported miRNAs. A map of RBPs and their binding targets from ENCODE eCLIP data [114,115], computational prediction [116], or enriched sequence motifs should further confirm the connection.

## 5. Conclusions

We analyzed the overall quality of all the publicly available small RNA-seq data on EVs. We found that small RNA reads in EVs were mainly derived from miRNAs, Y RNAs, and fragments of rRNA and tRNA. We further evaluated the technical and potential biological biases introduced by different EV isolation techniques in terms of small RNA proportions. The analysis presents a global picture of quality in EV small RNA-seq experiments and guides future quality control and selection of EV isolation methods.

We further studied the preferential enrichment of small RNA biotypes in EVs compared to their cellular levels. We revealed that miRNAs and snoRNA fragments are more likely to be retained in cells, while rRNA fragments and tRNA fragments are more likely to be transported into EVs. Although several miRNAs showed an overall preference to be either loaded into EVs or retained in cells, selective sorting of miRNAs into EVs was mostly study-specific. Together, our findings shed light on the preferential loading of specific RNA biotypes into EVs.

## Figures and Tables

**Figure 1 cancers-15-03446-f001:**
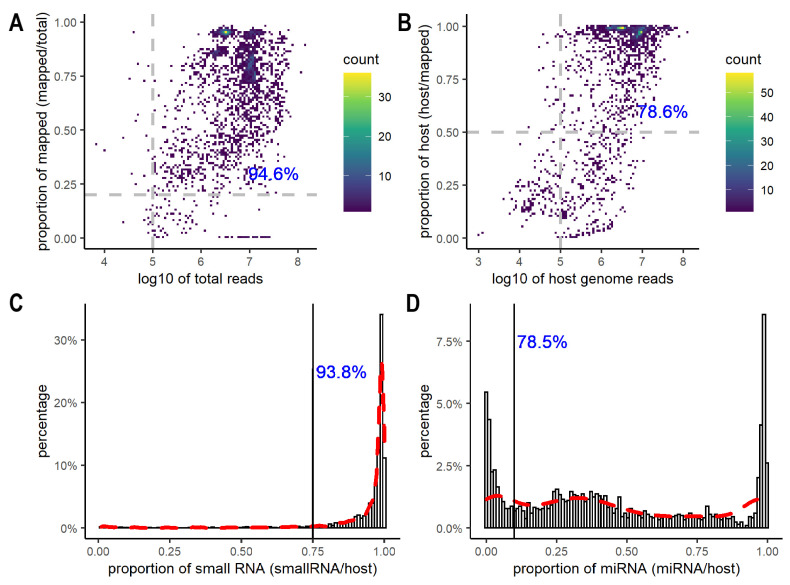
Quality control for EV small RNA-seq datasets. (**A**) Heatscatter plot showing the total number of reads on the *x*-axis and the total percentage of mapped reads on the *y*-axis. A total of 94.6% of datasets had more than 100,000 reads with greater than 20% mapping rates. (**B**) Heatscatter plot showing the total number of reads mapped to host genomes on the *x*-axis and percentage mapping rates to host genomes on the *y*-axis. A total of 78.6% of datasets had more than 100,000 reads that mapped to host genomes with greater than 50% being host genome reads. (**C**) Distribution of the proportion of host small RNAs. A total of 93.8% of the datasets reported small RNAs at greater than 75% of the total number of host genome reads. (**D**) Distribution of the proportion of the host miRNA. A total of 78.5% of the datasets reported miRNA content at greater than 10% of the total number of host genome reads.

**Figure 2 cancers-15-03446-f002:**
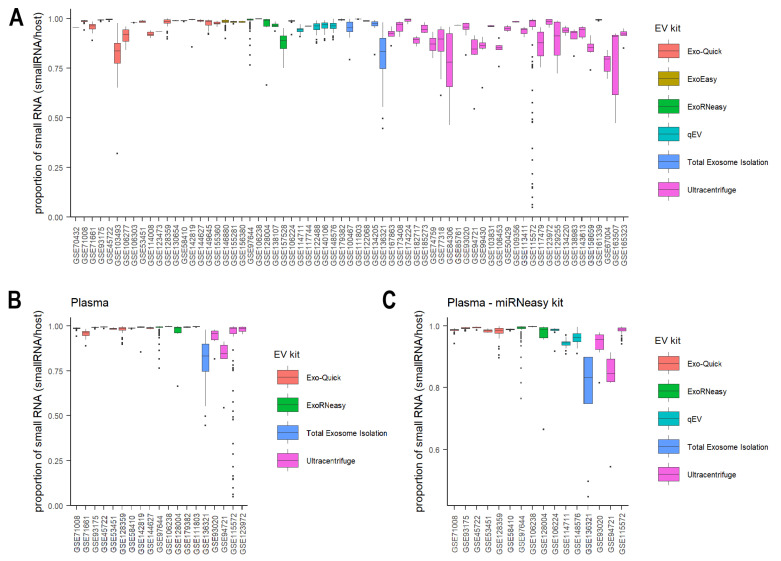
The proportion of host small RNAs across EV studies. Boxplot of the proportion of host small RNAs for all studies (**A**), studies from plasma (**B**), and studies from plasma using miRNeasy kits for RNA purification (**C**). Plots are colored by EV extraction methods.

**Figure 3 cancers-15-03446-f003:**
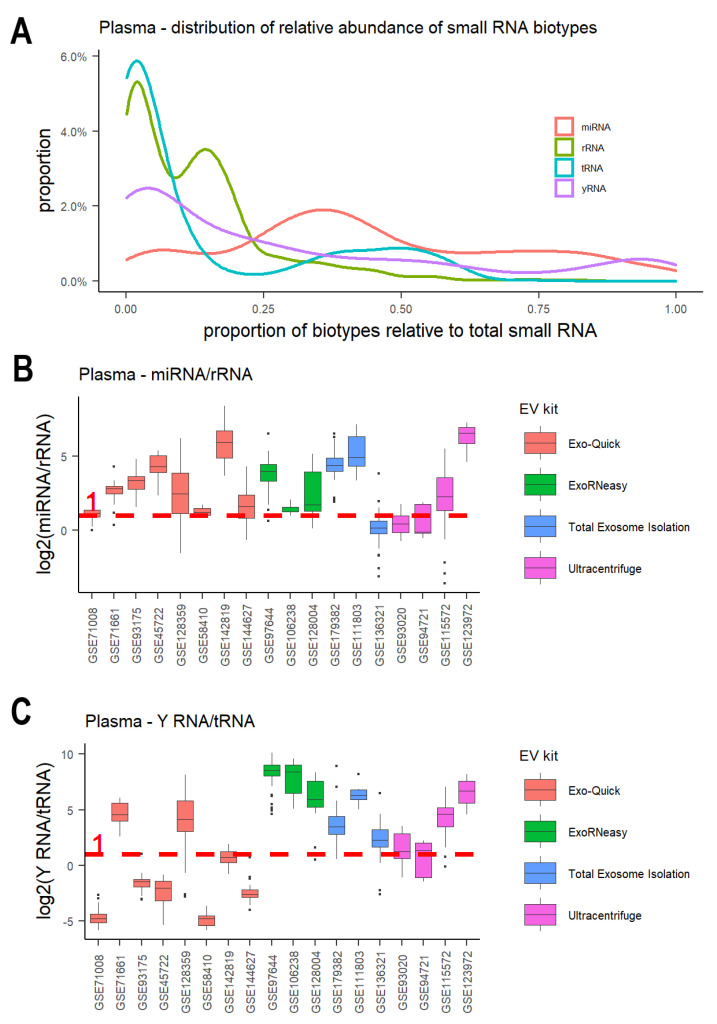
Relative abundance of small RNA biotypes across EV studies from plasma. (**A**) Distribution of the proportion of host miRNAs, rRNA fragments, tRNA fragments, and Y RNA fragments relative to total host small RNA. (**B**) Boxplot of log2 transformation of miRNA/rRNA ratio. (**C**) Boxplot of log2 transformation of Y RNA/tRNA ratio. The log2 ratio of 1 is denoted by the red dashed line. Plots are colored by EV extraction methods.

**Figure 4 cancers-15-03446-f004:**
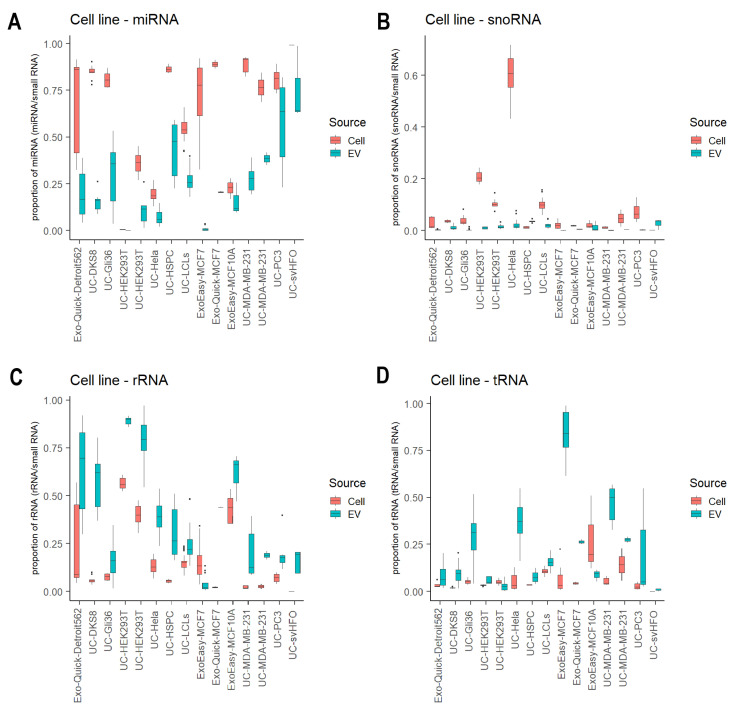
Enrichment of small RNA biotypes in EVs. Boxplot of the proportion of miRNAs (**A**), snoRNA fragments (**B**), rRNA fragments (**C**) and tRNA fragments (**D**) in EVs compared to their matched donor cellular levels.

**Figure 5 cancers-15-03446-f005:**
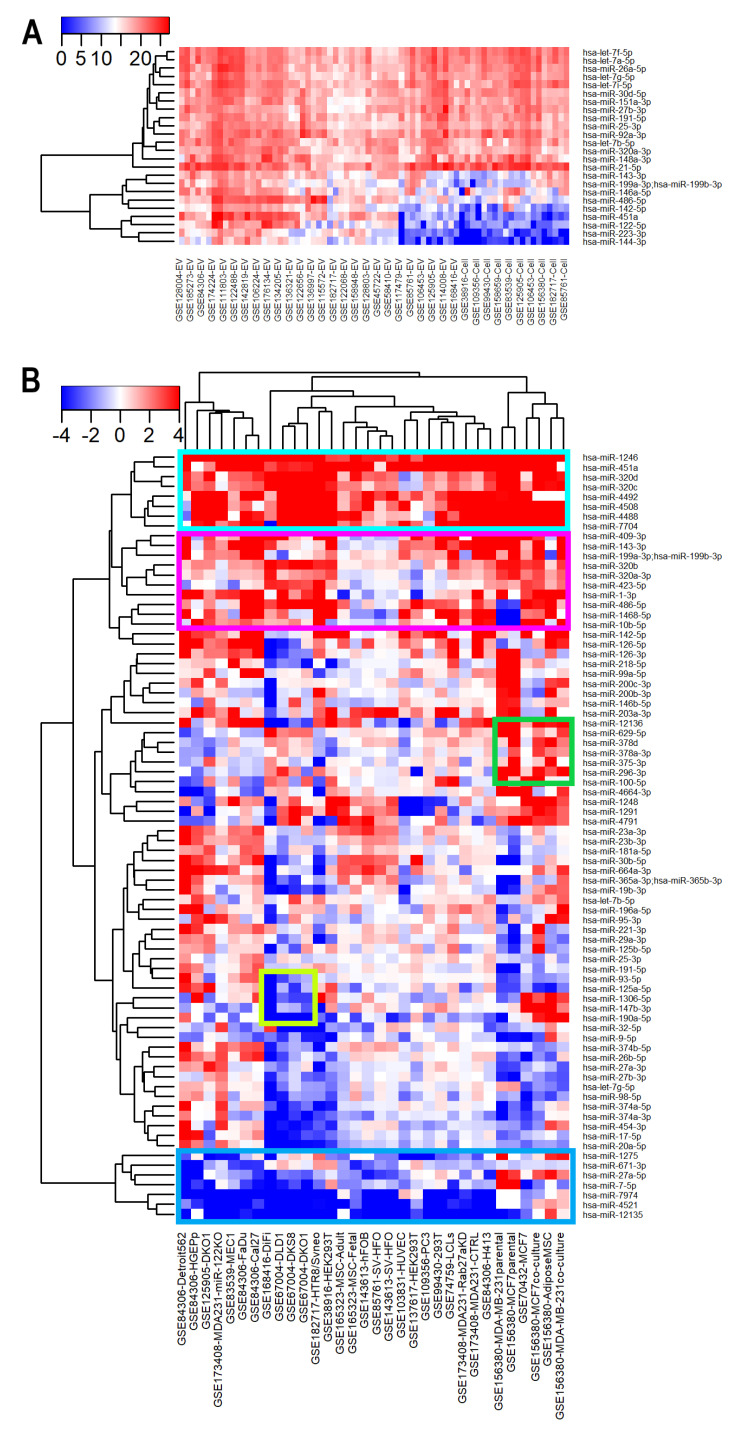
Enrichment of miRNAs in EVs. (**A**) Heatmap of the most abundant miRNAs in EV and/or donors. The expression of miRNAs was normalized to the median value across all samples. (**B**) Heatmap of log2 fold change in miRNAs expression between EVs and their matched donor cells.

## Data Availability

The data presented in this study are available in Appendix A.

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
