# Peer review of "Systematic Assessment of Small RNA Profiling in Human Extracellular Vesicles"

_cancers, 2023, doi:10.3390/cancers15133446_

Round 1

Reviewer 1 Report

The manuscript “Systematic assessment of small RNA profiling in human extracellular vesicles” by Wang et al. is a needed attempt to analyze smallRNA datasets in larger scale from EV isolates to understand which species are generally enriched in EV. This includes comparison to donor cells and analysis of robustness of expression from different EV isolation methods. Otherwise, this type of work is required in the EV field and I think it will find the interested readers if published, but I see one main problem. The authors do not take into account the compositions of the EV isolates – EV isolates have their characteristic compositions i.e. they are usually enriched in EV and to a different degree in other smallRNA carriers that vary according to the isolation principle used. I have pointed out below several parts relating to this and which should be reconsidered; I also suggest some changes to address this common problem (in the EV field) in the current study.  

-“To ensure EVs are enriched, we only kept studies where EVs were isolated using either commercial exosome extraction kits, such as Exo-Quick, ExoEasy, qEV, and total exosome isolation kit (Thermo), or

differential ultracentrifugation, which have been proved to isolate high-yield and high-purity EVs [40, 46-49].”

It is well known that each of these methods is prone to isolate some contaminants e.g. proteins by PEG-precipitation (Exo-Quick and TEI). To put it bluntly, one could almost claim that PEG-based methods are great in concentrating e.g. plasma samples as such. This point relates to the interpretation of results i.e. assigning the different smallRNA species to be sorted/secreted in EVs. I would highly recommend to correct this claim throughout the manuscript and say that what the authors have analyzed is the presence of different smallRNA species in different EV-enriched isolates (or fractions) from different methods.

-“Comparing variability after restricting to these studies came to the same conclusion that differential ultracentrifugation purification consistently showed higher variability and lower replicability in small RNA proportions within each study (Figs. 2B and 2C). These findings suggest that differential ultracentrifugation may not be as reproducible as other methods for EV isolation, which is consistent with previous findings [61, 62].”

However, if the other isolation methods are just reproducible ways of e.g. concentrating the body fluid almost as such (like PEG) or including a certain proportion of lipoproteins (SEC from plasma), then concluding that they are better in EV isolation is not entirely correct. They can be more reproducible in what they isolate, but it may not only (or even, not significantly) relate to isolating EV in a reproducible manner.

-“These findings suggest that miRNAs and snoRNA fragments are more likely to be retained in cells rather than to be transported into EVs. In contrast, rRNA fragments and tRNA fragments showed significantly higher levels in EVs compared to their cellular levels (p<0.01, paired t-test; Figs. 4C and Fig. 4D), suggesting they are more likely to be transported into EVs.”

Comparison of EVs to donor cells is not complete to make this kind of conclusions as the compositions of different EV isolates varies (EVs + variable amount of other smallRNA carriers) thus it is unclear where the smallRNAs are in the isolate.

- “In addition, we discovered nine miRNAs highly abundant in EVs but not in cells, including hsa-miR-451a, hsa-miR-486-5p, hsa-miR-122-5p, hsa-miR-146a-5p, hsa- miR-199a-3p;hsa-miR-199b-3p, hsa-miR-143-3p, hsa-miR-144-3p, hsa-miR-21-5p, and hsa- miR-223-5p (Fig. 5A). Among them, mir-451a and miR-144 have been previously reported to show higher levels in human EVs compared to cellular levels [32].”

MiRs 486-5p and 451a are among the most abundant miRs in blood plasma, generally. Therefore, they could be also abundant in EV samples just because the isolation methods usually copurify certain amounts of “contaminants” (other secreted carriers of miRNA).

Discussion

-The discussion is thoroughly discussing the effect of EV heterogeneity related aspects and its effects on the smallRNAs in EVs – however – it misses to discuss the contributions of “contaminants” meaning other secreted material carrying smallRNA. I suggest that the authors discuss this topic and dedicate the manuscript to identify the most robust methods in terms of reproducibility of isolating smallRNAs but not claiming this relates to the reproducibility of isolating EVs. For UC, the greater variability could actually come from the fact that UC succeeds to purify the EV better than e.g. PEG, which robustly precipitates almost everything in the sample. However, UC has more equipment, protocol and operator dependent variation and hence the variability.

Other comments

-GEO abbreviation has not been explained

-“ We decided to exclude experimental series where more than half of samples had either < 75% small RNAs or < 10% miRNA reads.”

Why?

Reviewer 2 Report

In this article, Wang and colleagues provide guidance on quality control and the selection of EV isolation methods and enhance the interpretation of small RNA contents and preferential loading in EVs

Introduction: A point that could be improved is related to the introductive part of the paper  addressing the need for new technologies for the association of a specific marker with an exosome subtype and the exosome subtype to a particular function and/or group of functions s (PMID: 36972680, PMID: 37012233, PMID: 36310768 or others)

Methods: The methodology section is well described and doesn't need particular revision even if more details in order the exosome purification protocol would be very useful.

Results: The study was well conducted and the results are clearly demonstrated. Figure legends are quite informative and figure resolution looks appropriate. The quality of purified EVs needs to be increased including nanosizer analysis and western blotting for CD63, CD81 and others.

The overall consideration goes in the direction of a really good paper and I feel that also taking into consideration the above comments the paper could be appreciated by the frontiers readers. Good luck.

Moderate editing of English language required

Round 2

Reviewer 1 Report

The authors have done changes according to my suggestions - the interpretations and the manuscript have improved.